# ON DIFFERENTIALLY PRIVATE STRING DISTANCES

## ABSTRACT

Given a database of bit strings $A_1, \ldots, A_m \in \{0,1\}^n$, a fundamental data structure task is to estimate the distances between a given query $B \in \{0,1\}^n$ with all the strings in the database. In addition, one might further want to ensure the integrity of the database by releasing these distance statistics in a secure manner. In this work, we propose differentially private (DP) data structures for this type of tasks, with a focus on Hamming and edit distance. On top of the strong privacy guarantees, our data structures are also time- and space-efficient. In particular, our data structure is $\epsilon$-DP against any sequence of queries of arbitrary length, and for any query $B$ such that the maximum distance to any string in the database is at most $k$, we output $m$ distance estimates. Moreover,

- For Hamming distance, our data structure answers any query in $\widetilde{O}(mk + n)$ time and each estimate deviates from the true distance by at most $\widetilde{O}(k/e^{\epsilon/\log k})$;

- For edit distance, our data structure answers any query in $\widetilde{O}(mk^2 + n)$ time and each estimate deviates from the true distance by at most $\widetilde{O}(k/e^{\epsilon/(\log k \log n)})$.

For moderate $k$, both data structures support sublinear query operations. We obtain these results via a novel adaptation of the randomized response technique as a bit flipping procedure, applied to the sketched strings.

## 1 INTRODUCTION

Estimating string distances is one of the most fundamental problems in computer science and information theory, with rich applications in high-dimensional geometry, computational biology and machine learning. The problem could be generically formulated as follows: given a collection of strings $A_1, \ldots, A_m \in \Sigma^n$ where $\Sigma$ is the alphabet, the goal is to design a data structure to preprocess these strings such that when a query $B \in \Sigma^n$ is given, the data structure needs to quickly output estimates of $\|A_i - B\|$ for all $i \in [m]$, where $\|\cdot\|$ is the distance of interest. Assuming the symbols in $\Sigma$ can allow constant time access and operations, a naïve implementation would be to simply compute all the distances between $A_i$'s and $B$, which would require $O(mn)$ time. Designing data structures with $o(mn)$ query time has been the driving research direction in string distance estimations. To make the discussion concrete, in this work we will focus on binary alphabet ($\Sigma = \{0, 1\}$) and for distance, we will study Hamming and edit distance. Hamming distance (Hamming, 1950) is one of the most natural distance measurements for binary strings, with its deep root in error detecting and correction for codes. It finds large array of applications in database similarity searches (Indyk & Motwani, 1998; Charikar, 2002; Norouzi et al., 2012) and clustering algorithms (Huang, 1997; Huang & Ng, 1999).

Compared to Hamming distance, edit distance or the Levenshtein distance (Levenshtein, 1966) could be viewed as a more robust distance measurement for strings: it counts the minimum number of operations (including insertion, deletion and substitution) to transform from $A_i$ to $B$. To see the robustness compared to Hamming distance, consider $A_i = (01)^n$ and $B = (10)^n$, the Hamming distance between these two strings is $n$, but $A_i$ could be easily transformed to $B$ by deleting the first bit and adding a 0 to the end, yielding an edit distance of 2. Due to its flexibility, edit distance is particularly useful for sequence alignment in computational biology (Wang et al., 2015; Young et al., 2021; Berger et al., 2021), measuring text similarity (Navarro, 2001; Sidorov et al., 2015) and natural language processing, speech recognition (Fiscus et al., 2006; Droppo & Acero, 2010) and time series analysis (Marteau, 2009; Gold & Sharir, 2018).

In addition to data structures with fast query times, another important consideration is to ensure the database is *secure*. Consider the scenario where the database consists of private medical data of $m$ patients, where each of the $A_i$ is the characteristic vector of $n$ different symptoms. A malicious adversary might attempt to count the number of symptoms each patient has by querying $\mathbf{0}_n$, or detecting whether patient $i$ has symptom $j$ by querying $e_j$ and $\mathbf{0}_n$ where $e_j$ is the $j$-th standard basis in $\mathbb{R}^n$. It is hence crucial to curate a private scheme so that the adversary cannot distinguish the case whether the patient has symptom $j$ or not. This notion of privacy has been precisely captured by *differential privacy* (Dwork, 2006; Dwork et al., 2006), which states that for neighboring databases[1], the output distribution of the data structure query should be close with high probability, hence any adversary cannot distinguishable between the two cases.

Motivated by both privacy and efficiency concerns, we ask the following natural question:

> *Is it possible to design a data structures to estimate Hamming and edit distance, that are both differentially private, and time/space-efficient?*

We provide an affirmative answer to the above question, with the main results summarized in the following two theorems. We will use $D_{\mathrm{ham}}(A, B)$ to denote the Hamming distance between $A$ and $B$, and $D_{\mathrm{edit}}(A, B)$ to denote the edit distance between $A$ and $B$. We also say a data structure is $\epsilon$-DP if it provides $\epsilon$-DP outputs against any sequence of queries, of arbitrary length.

**Theorem 1.1.** *Let $A_1, \ldots, A_m \in \{0, 1\}^n$ be a database, $k \in [n]$ and $\epsilon > 0, \beta \in (0, 1)$, then there exists a randomized algorithm with the following guarantees:*

- *The data structure is $\epsilon$-DP;*

- *It perprocesses $A_1, \ldots, A_m$ in time $\widetilde{O}(mn)$ time[2];*

- *It consumes $\widetilde{O}(mk)$ space;*

- *Given any query $B \in \{0, 1\}^n$ such that $\max_{i \in [m]} D_{\mathrm{ham}}(A_i, B) \leq k$, it outputs $m$ estimates $z_1, \ldots, z_m$ with $|z_i - D_{\mathrm{ham}}(A_i, B)| \leq \widetilde{O}(k/e^{\epsilon/\log k})$ for all $i \in [m]$ in time $\widetilde{O}(mk + n)$, and it succeeds with probability at least $1 - \beta$.*

**Theorem 1.2.** *Let $A_1, \ldots, A_m \in \{0, 1\}^n$ be a database, $k \in [n]$ and $\epsilon > 0, \beta \in (0, 1)$, then there exists a randomized algorithm with the following guarantees:*

- *The data structure is $\epsilon$-DP;*

- *It perprocesses $A_1, \ldots, A_m$ in time $\widetilde{O}(mn)$ time;*

- *It consumes $\widetilde{O}(mn)$ space;*

- *Given any query $B \in \{0, 1\}^n$ such that $\max_{i \in [m]} D_{\mathrm{edit}}(A_i, B) \leq k$, it outputs $m$ estimates $z_1, \ldots, z_m$ with $|z_i - D_{\mathrm{edit}}(A_i, B)| \leq \widetilde{O}(k/e^{\epsilon/(\log k \log n)})$ for all $i \in [m]$ in time $\widetilde{O}(mk^2 + n)$, and it succeeds with probability at least $1 - \beta$.*

Before diving into the details, we would like to make several remarks regarding our data structure results. Note that instead of solving the exact Hamming distance and edit distance problem, we impose the assumption that the query $B$ has the property that for any $i \in [m]$, $\|A_i - B\| \leq k$. Such an assumption might seem restrictive at its first glance, but under the standard complexity assumption Strong Exponential Time Hypothesis (SETH) (Impagliazzo & Paturi, 2001; Impagliazzo et al., 2001), it is known that there is no $O(n^{2-o(1)})$ time algorithm exists for exact or even approximate edit distance (Belazzougui & Zhang, 2016; Chakraborty et al., 2016a;b; Naumovitz et al., 2017; Rubinstein et al., 2019; Rubinstein & Song, 2020; Goldenberg et al., 2020; Jin et al., 2021; Boroujeni et al., 2021; Kociumaka et al., 2021; Bhattacharya & Koucký, 2023; Koucký & Saks, 2024). It is therefore natural to impose assumptions that the query is "near" to the database in pursuit of faster algorithms (Ukkonen, 1985; Myers, 1986; Landau & Vishkin, 1988; Goldenberg et al., 2019;

---

[1]In our case, we say two database $\mathcal{D}_1$ and $\mathcal{D}_2$ are neighboring if there exists one $i \in [n]$ such that $\mathcal{D}_1(A_i)$ and $\mathcal{D}_2(A_i)$ differs by one bit.

[2]Throughout the paper, we will use $\widetilde{O}(\cdot)$ to suppress polylogarithmic factors in $m, n, k$ and $1/\beta$.

Kociumaka & Saha, 2020; Goldenberg et al., 2023). In fact, assuming SETH, $O(n + k^2)$ runtime for edit distance when $m = 1$ is optimal up to sub-polynomial factors (Goldenberg et al., 2023). Thus, in this paper, we consider the setting where $\max_{i \in [m]} \|A_i - B\| \le k$ for both Hamming and edit distance and show how to craft private and efficient mechanisms for this class of distance problems.

Regarding privacy guarantees, one might consider the following simple augmentation to any fast data structure for Hamming distance: compute the distance estimate via the data structure, and add Laplace noise to it. Since changing one coordinate of the database would lead to the Hamming distance change by at most 1, Laplace mechanism would properly handle this case. However, our goal is to release a *differentially private data structure* that is robust against potentially infinitely many queries, and a simple output perturbation won't be sufficient as an adversary could simply query with the same $B$, average them to reduce the variance and obtain a relatively accurate estimate of the de-noised output. To address this issue, we consider the *differentially private function release communication model* (Hall et al., 2013), where the curator releases an $\epsilon$-DP description of a function $\widehat{e}(\cdot)$ that is $\epsilon$-DP without seeing any query in advance. The client can then use $\widehat{e}(\cdot)$ to compute $\widehat{e}(B)$ for any query $B$. This strong guarantee ensures that the client could feed infinitely many queries to $\widehat{e}(\cdot)$ without compromising the privacy of the database.

## 2 RELATED WORK

**Differential Privacy.** Differential privacy is a ubiquitous notion for protecting the privacy of database. Dwork et al. (2006) first introduced this concept, which characterizes a class of algorithms such that when inputs are two neighboring datasets, with high probability the output distributions are similar. Differential privacy has a wide range of applications in general machine learning (Chaudhuri & Monteleoni, 2008; Williams & McSherry, 2010; Jayaraman & Evans, 2019; Triastcyn & Faltings, 2020), training deep neural networks (Abadi et al., 2016; Bagdasaryan et al., 2019), computer vision (Zhu et al., 2020; Luo et al., 2021; Torkzadehmahani et al., 2019), natural language processing (Yue et al., 2021; Weggenmann & Kerschbaum, 2018), large language models (Gao et al., 2023; Yu et al., 2022), label protect (Yang et al., 2022), multiple data release (Wu et al., 2022), federated learning (Sun et al., 2023; Song et al., 2023a) and peer review (Ding et al., 2022). In recent years, differential privacy has been playing an important role for data structure design, both in making these data structures robust against adaptive adversary (Beimel et al., 2022; Hassidim et al., 2022; Song et al., 2023b; Cherapanamjeri et al., 2023) and in the function release communication model (Hall et al., 2013; Huang & Roth, 2014; Wang et al., 2016; Aldà & Rubinstein, 2017; Coleman & Shrivastava, 2021; Wagner et al., 2023; Backurs et al., 2024).

**Hamming Distance and Edit Distance.** Given bit strings $A$ and $B$, many distance measurements have been proposed that capture various characteristics of bit strings. Hamming distance was first studied by Hamming (Hamming, 1950) in the context of error correction for codes. From an algorithmic perspective, Hamming distance is mostly studied in the context of approximate nearest-neighbor search and locality-sensitive hashing (Indyk & Motwani, 1998; Charikar, 2002). When it is known that the query $B$ has the property $D_{\mathrm{ham}}(A, B) \le k$, Porat & Lipsky (2007) shows how to construct a sketch of size $\widetilde{O}(k)$ in $\widetilde{O}(n)$ time, and with high probability, these sketches preserve Hamming distance. Edit distance, proposed by Levenshtein (Levenshtein, 1966), is a more robust notion of distance between bit strings. It has applications in computational biology (Wang et al., 2015; Young et al., 2021; Berger et al., 2021), text similarity (Navarro, 2001; Sidorov et al., 2015) and speech recognition (Fiscus et al., 2006; Droppo & Acero, 2010). From a computational perspective, it is known that under the Strong Exponential Time Hypothesis (SETH), no algorithm can solve edit distance in $O(n^{2-o(1)})$ time, even its approximate variants (Belazzougui & Zhang, 2016; Chakraborty et al., 2016a;b; Naumovitz et al., 2017; Rubinstein et al., 2019; Rubinstein & Song, 2020; Goldenberg et al., 2020; Jin et al., 2021; Boroujeni et al., 2021; Kociumaka et al., 2021; Bhattacharya & Koucký, 2023; Koucký & Saks, 2024). Hence, various assumptions have been imposed to enable more efficient algorithm design. The most related assumption to us is that $D_{\mathrm{edit}}(A, B) \le k$, and in this regime various algorithms have been proposed (Ukkonen, 1985; Myers, 1986; Landau & Vishkin, 1988; Goldenberg et al., 2019; Kociumaka & Saha, 2020; Goldenberg et al., 2023). Under SETH, it has been shown that the optimal dependence on $n$ and $k$ is $O(n + k^2)$, up to sub-polynomial factors (Goldenberg et al., 2023).

## 3 PRELIMINARY

Let $E$ be an event, we use $\mathbf{1}[E]$ to denote the indicator variable if $E$ is true. Given two length-$n$ bit strings $A$ and $B$, we use $D_{\mathrm{ham}}(A, B)$ to denote $\sum_{i=1}^{n} \mathbf{1}[A_i = B_i]$. We use $D_{\mathrm{edit}}(A, B)$ to denote the edit distance between $A$ and $B$, i.e., the minimum number of operations to transform $A$ to $B$ where the allowed operations are insertion, deletion and substitution. We use $\oplus$ to denote the XOR operation. For any positive integer $n$, we use $[n]$ to denote the set $\{1, 2, \cdots, n\}$. We use $\Pr[\cdot]$, $\mathbb{E}[\cdot]$ and $\mathrm{Var}[\cdot]$ to denote probability, expectation and variance respectively.

### 3.1 CONCENTRATION BOUNDS

We will mainly use two concentration inequalities in this paper.

**Lemma 3.1** (Chebyshev's Inequality). *Let $X$ be a random variable with $0 < Var[X] < \infty$. For any real number $t > 0$,*

$$\Pr[|X - \mathbb{E}[X]| > t] \leq \frac{Var[X]}{t^2}.$$

**Lemma 3.2** (Hoeffding's Inequality). *Let $X_1, \ldots, X_n$ with $a_i \leq X_i \leq b_i$ almost surely. Let $S_n = \sum_{i=1}^{n} X_i$, then for any real number $t > 0$,*

$$\Pr[|S_n - \mathbb{E}[S_n]| > t] \leq 2 \exp\left(-\frac{2t^2}{\sum_{i=1}^{n}(b_i - a_i)^2}\right).$$

### 3.2 DIFFERENTIAL PRIVACY

Differential privacy (DP) is the key privacy measure we will be trying to craft our algorithm to possess it. In this paper, we will solely focus on pure DP ($\epsilon$-DP).

**Definition 3.3** ($\epsilon$-Differential Privacy). *We say an algorithm $\mathcal{A}$ is $\epsilon$-differentially private ($\epsilon$-DP) if for any two neighboring databases $\mathcal{D}_1$ and $\mathcal{D}_2$ and any subsets of possible outputs $S$, we have*

$$\Pr[\mathcal{A}(\mathcal{D}_1) \in S] \leq e^{\epsilon} \cdot \Pr[\mathcal{A}(\mathcal{D}_2) \in S],$$

*where the probability is taken over the randomness of $\mathcal{A}$.*

Since we will be designing data structures, we will work with the *function release communication model* (Hall et al., 2013) where the goal is to release a function that is $\epsilon$-DP against any sequence of queries of arbitrary length.

**Definition 3.4** ($\epsilon$-DP Data Structure). *We say a data structure $\mathcal{A}$ is $\epsilon$-DP, if $\mathcal{A}$ is $\epsilon$-DP against any sequence of queries of arbitrary length. In other words, the curator will release an $\epsilon$-DP description of a function $\widehat{e}(\cdot)$ without seeing any query in advance.*

Finally, we will be utilizing the *post-processing* property of $\epsilon$-DP.

**Lemma 3.5** (Post-Processing). *Let $\mathcal{A}$ be $\epsilon$-DP, then for any deterministic or randomized function $g$ that only depends on the output of $\mathcal{A}$, $g \circ \mathcal{A}$ is also $\epsilon$-DP.*

## 4 DIFFERENTIALLY PRIVATE HAMMING DISTANCE DATA STRUCTURE

To start off, we introduce our data structure for differentially private Hamming distance. In particular, we will adapt a data structure due to Porat & Lipsky (2007): this data structure computes a sketch of length $\widetilde{O}(k)$ bit string to both the database and query, then with high probability, one could retrieve the Hamming distance from these sketches. Since the resulting sketch is also a bit string, a natural idea is to inject Laplace noise on each coordinate of the sketch. Since for two neighboring databases, only one coordinate would change, we could add Laplace noise of scale $1/\epsilon$ to achieve $\epsilon$-DP. However, this approach has a critical issue: one could show that with high probability, the magnitude of each noise is roughly $O(\epsilon^{-1} \log k)$, aggregating the $k$ coordinates of the sketch, this leads to a total error of $O(\epsilon^{-1} k \log k)$. To decrease this error to $O(1)$, one would have to choose $\epsilon = k \log k$, which is too large for most applications.

Instead of Laplace noises, we present a novel scheme that flips each bit of the sketch with certain probability. Our main contribution is to show that this simple scheme, while produces a biased estimator, the error is only $O(e^{-\epsilon/\log k}k)$. Let $t = \log k/\epsilon$, we see that the Laplace mechanism has an error of $O(t^{-1}k)$ and our error is only $O(e^{-t}k)$, which is exponentially small! In what follows, we will describe a data structure when the database is only one string $A$ and with constant success probability, and we will discuss how to extend it to $m$ bit strings, and how to boost the success probability to $1 - \beta$ for any $\beta > 0$. We summarize the main result below.

**Theorem 4.1.** *Given a string $A$ of length $n$. There exists an $\epsilon$-DP data structure* DPHAMMINGDIS-TANCE *(Algorithm 1), with the following operations*

- INIT($A \in \{0,1\}^n$): *It takes a string $A$ as input. This procedure takes $O(n\log k + k\log^3 k)$ time.*

- QUERY($B \in \{0,1\}^n$): *for any $B$ with $z := D_{\mathrm{ham}}(A, B) \leq k$, QUERY($B$) outputs a value $\widetilde{z}$ such that $|\widetilde{z} - z| = \widetilde{O}(k/e^{\epsilon/\log k})$ with probability $0.99$, and the result is $\epsilon$-DP. This procedure takes $O(n\log k + k\log^3 k)$ time.*

---

**Algorithm 1** Differential Private Hamming Distance Query

---

1: **data structure** DPHAMMINGDISTANCE $\qquad\qquad\qquad\qquad\qquad\qquad\qquad$ ▷ Theorem 4.1
2: **members**
3: $\qquad M_1, M_2, M_3 \in \mathbb{N}_+$
4: $\qquad h(x) : [2n] \to [M_2]$ $\qquad\qquad\qquad\qquad$ ▷ $h$ and $g$ are public random hash function
5: $\qquad g(x, i) : [2n] \times [M_1] \to [M_3]$
6: $\qquad S_{i,j,c} \in \{0,1\}^{M_1 \times M_2 \times M_3}$ for all $i \in [M_1], j \in [M_2], c \in [M_3]$ $\qquad$ ▷ $S$ represents the sketch
7: **end members**
8:
9: **procedure** ENCODE($A \in \{0,1\}^n, n$) $\qquad\qquad\qquad\qquad\qquad\qquad\qquad\qquad$ ▷ Lemma 4.2
10: $\qquad S^*_{i,j,c} \leftarrow 0$ for all $i, j, c$
11: $\qquad$ **for** $p \in [n]$ **do**
12: $\qquad\qquad$ **for** $i \in [M_1]$ **do**
13: $\qquad\qquad\qquad j \leftarrow h(2(p-1) + A_p)$
14: $\qquad\qquad\qquad c \leftarrow g(2(p-1) + A_p, i)$
15: $\qquad\qquad\qquad S^*_{i,j,c} \leftarrow S^*_{i,j,c} \oplus 1$
16: $\qquad\qquad$ **end for**
17: $\qquad$ **end for**
18: $\qquad$ **return** $S^*$
19: **end procedure**
20:
21: **procedure** INIT($A \in \{0,1\}^n, n \in \mathbb{N}_+, k \in \mathbb{N}_+, \epsilon' \in \mathbb{R}_+$) $\qquad\qquad\qquad\qquad$ ▷ Lemma 4.3
22: $\qquad M_1 \leftarrow 10\log k$
23: $\qquad M_2 \leftarrow 2k$
24: $\qquad M_3 \leftarrow 400\log^2 k$
25: $\qquad S \leftarrow$ ENCODE($A, n$)
26: $\qquad$ Flip each $S_{i,j,c}$ with independent probability $1/(1 + e^{\epsilon'/(2M_1)})$
27: **end procedure**
28:
29: **procedure** QUERY($B \in \{0,1\}^n$) $\qquad\qquad\qquad\qquad\qquad\qquad\qquad\qquad\qquad$ ▷ Lemma 4.7
30: $\qquad S^B \leftarrow$ ENCODE($B, n$)
31: $\qquad$ **return** $0.5 \cdot \sum_{j=1}^{M_2} \max_{i \in [M_1]}(\sum_{c=1}^{M_3} |S_{i,j,c} - S^B_{i,j,c}|)$
32: **end procedure**
33: **end data structure**

---

To achieve the results above, we set parameters $M_1 = O(\log k), M_2 = O(k), M_3 = O(\log^2 k)$ in Algorithm 1.

We divide the proof of Theorem 4.1 into the following subsections:

## 4.1 TIME COMPLEXITY

Note that both the initializing and query run ENCODE (Algorithm 1) exactly once, we show that the running time of ENCODE is $O(n \log k)$.

**Lemma 4.2.** *Given $M_1 = O(\log k)$, the running time of* ENCODE *(Algorithm 1) is $O(n \log k)$.*

*Proof.* In ENCODE, for each character in the input string, the algorithm iterate $M_1$ times. Therefore the total time complexity is $O(n \cdot M_1) = O(n \log k)$. □

## 4.2 PRIVACY GUARANTEE

Next we prove our data structure is $\epsilon$-DP.

**Lemma 4.3.** *Let $A$ and $A'$ be two strings that differ on only one position. Let $\mathcal{A}(A)$ and $\mathcal{A}(A')$ be the output of* INIT *(Algorithm 1) given $A$ and $A'$. For any output $S$, we have:*

$$\Pr[\mathcal{A}(A) = S] \leq e^\epsilon \cdot \Pr[\mathcal{A}(A') = S].$$

Due to space limitation, we defer the proof to Appendix A.

## 4.3 UTILITY GUARANTEE

The utility analysis is much more involved than privacy and runtime analysis. We defer the proofs to the appendix, while stating key lemmas.

We first consider the distance between sketches of $A$ and $B$ without the random flipping process. Let $E(A), E(B)$ be ENCODE$(A)$ and ENCODE$(B)$. We prove with probability 0.99, $D_{\mathrm{ham}}(A, B) = 0.5 \cdot \sum_{j=1}^{M_2} \max_{i \in [M_1]} (\sum_{c=1}^{M_3} |E(A)_{i,j,c} - E(B)_{i,j,c}|)$. Before we present the error guarantee, we will first introduce two technical lemmas. If we let $T = \{p \subseteq [n] \mid A_p \neq B_p\}$ denote the set of "bad" coordinates, then for each coordinate in the sketch, it only contains a few bad coordinates.

**Lemma 4.4.** *Define set $T := \{p \in [n] \mid A_p \neq B_p\}$. Define set $T_j := \{p \subseteq T \mid h(p) = j\}$. When $M_2 = 2k$, with probability 0.99, for all $j \in [M_2]$, we have $|T_j| \leq 10 \log k$, i.e.,*

$$\Pr[\forall j \in [M_2], \ | \ |T_j| \leq 10 \log k] \geq 0.99.$$

The next lemma shows that with high probability, the second level hashing $g$ will hash bad coordinates to distinct buckets.

**Lemma 4.5.** *When $M_1 = 10 \log k, M_2 = 2k, M_3 = 400 \log^2 k$, with probability 0.98, for all $j \in [M_2]$, there is at least one $i \in [M_1]$, such that all values in $\{g(2(p-1) + A_p, i) \mid p \in T_j\} \bigcup \{g(2(p-1) + B_p, i) \mid p \in T_j\}$ are distinct.*

With these two lemmas in hand, we are in the position to prove the error bound before the random bit flipping process.

**Lemma 4.6.** *Let $E(A), E(B)$ be the output of* ENCODE$(A)$ *and* ENCODE$(B)$. *With probability 0.98, $D_{\mathrm{ham}}(A, B) = 0.5 \cdot \sum_{j=1}^{M_2} \max_{i \in [M_1]} (\sum_{c=1}^{M_3} |E(A)_{i,j,c} - E(B)_{i,j,c}|)$.*

Our final result provides utility guarantees for Algorithm 1.

**Lemma 4.7.** *Let $z$ be $D_{ham}(A, B)$, $\widetilde{z}$ be the output of* QUERY$(B)$*(Algorithm 1). With probability 0.98, $|z - \widetilde{z}| \leq O(k \log^3 k / e^{\epsilon / \log k})$.*

*Proof.* From Lemma 4.6, we know with probability 0.98, when $\epsilon \to \infty$ (i.e. without the random flip process), the output of QUERY$(B)$ (Algorithm 1) equals the exact hamming distance.

We view the random flip process as random variables. Let random variables $R_{i,j,c}$ be 1 with probability $1/(1 + e^{\epsilon/M_1})$, or 0 with probability $1 - 1/(1 + e^{\epsilon/M_1})$. So we have

$$|\widetilde{z} - z| = \sum_{j=1}^{M_2} \max_{i \in [M_1]} (\sum_{c=1}^{M_3} R_{i,j,c})$$

$$\leq \sum_{j=1}^{M_2} \sum_{i=1}^{M_1} (\sum_{c=1}^{M_3} R_{i,j,c}),$$

where the second step follows from $\max_i \leq \sum_i$ when all the summands are non-negative.

Therefore, the expectation of $\widetilde{z} - z$ is:

$$\mathbb{E}[|\widetilde{z} - z|] = M_1 M_2 M_3 \cdot \mathbb{E}[R_{i,j,c}]$$
$$= k \log^3 k \cdot \frac{1}{(1 + e^{\epsilon/\log k})}$$
$$\leq O(\frac{k \log^3 k}{e^{\epsilon/\log k}}),$$

where the last step follows from simple algebra. The variance of $\widetilde{z} - z$ is:

$$\text{Var}[|\widetilde{z} - z|] = M_1 M_2 M_3 \cdot \text{Var}[R_{i,j,c}]$$
$$= k \log^3 k \cdot \frac{1}{(1 + e^{\epsilon/\log k})} \cdot (1 - \frac{1}{(1 + e^{\epsilon/\log k})}).$$

Using Chebyshev's inequality (Lemma 3.1), we have

$$\Pr[|\widetilde{z} - z| \geq O(\frac{k \log^3 k}{e^{\epsilon/\log k}})] \leq 0.01.$$

Thus we complete the proof. $\qquad\square$

**Remark 4.8.** *We will show how to generalize Theorem 4.1 to $m$ bit strings, and how to boost the success probability to $1 - \beta$. To boost the success probability, we note that individual data structure succeeds with probability 0.99, we could take $\log(1/\beta)$ independent copies of the data structure, and query all of them. By a standard Chernoff bound argument, with probability at least $1 - \beta$, at least $3/4$ fraction of these data structures would output the correct answer, hence what we could do is to take the median of these answers. These operations blow up both INIT and QUERY by a factor of $\log(1/\beta)$ in its runtime. Generalizing for a database of $m$ strings is relatively straightforward: we will run the INIT procedure to $A_1, \ldots, A_m$, this would take $O(mn \log k + mk \log^3 k)$ time. For each query, note we only need to ENCODE the query once, and we can subsequently compute the Hamming distance from the sketch for $m$ sketched database strings, therefore the total time for query is $O(n \log k + mk \log^3 k)$. It is important to note that as long as $k \log^3 k < n$, the query time is sublinear. Finally, we could use the success probability boosting technique described before, that uses $\log(m/\beta)$ data structures to account for a union bound over the success of all distance estimates.*

## 5 DIFFERENTIALLY PRIVATE EDIT DISTANCE DATA STRUCTURE

Our algorithm for edit distance follows from the dynamic programming method introduced by Ukkonen (1985); Landau et al. (1998); Landau & Vishkin (1988); Myers (1986). We note that a key procedure in these algorithms is a subroutine to estimate *longest common prefix* (LCP) between two strings $A$ and $B$ and their substrings. We design an $\epsilon$-DP data structure for LCP based on our $\epsilon$-DP Hamming distance data structure. Due to space limitation, we defer the details of the DP-LCP data structure to Appendix B. In the following discussion, we will assume access to a DP-LCP data structure with the following guarantees:

**Theorem 5.1.** *Given a string $A$ of length $n$. There exists an $\epsilon$-DP data structure DPLCP (Algorithm 3 and Algorithm 4) supporting the following operations*

- INIT($A \in \{0,1\}^n$): *It preprocesses an input string $A$. This procedure takes $O(n(\log k + \log \log n))$ time.*

- INITQUERY($B \in \{0,1\}^n$): *It preprocesses an input query string $B$. This procedure take $O(n(\log k + \log \log n))$ time.*

- QUERY($i, j$): *Let $w$ be the longest common prefix of $A[i : n]$ and $B[j : n]$ and $\widetilde{w}$ be the output of QUERY($i, j$), With probability $1 - 1/(300k^2)$, we have: 1). $\widetilde{w} \geq w$; 2). $\mathbb{E}[D_{\text{ham}}(A[i : i+\widetilde{w}], B[j : j+\widetilde{w}])] \leq O((\log k + \log \log n)/e^{\epsilon/(\log k \log n)})$. This procedure takes $O(\log^2 n(\log k + \log \log n))$ time.*

We will be basing our edit distance data structure on the following result, which achieves the optimal dependence on $n$ and $k$ assuming SETH:

**Lemma 5.2.** *(Landau et al., 1998) Given two strings $A$ and $B$ of length $n$. If the edit distance between $A$ and $B$ is no more than $k$, there is an algorithm which computes the edit distance between $A$ and $B$ in time $O(k^2 + n)$.*

We start from a naïve dynamic programming approach. Define $D(i, j)$ to be the edit distance between string $A[1 : i]$ and $B[1 : j]$. We could try to match $A[i]$ and $B[j]$ by inserting, deleting and substituting, which yields the following recurrence:

$$D(i, j) = \min \begin{cases} D(i-1, j) + 1 & \text{if } i > 0; \\ D(i-1, j-1) + 1 & \text{if } j > 0; \\ D(i-1, j-1) + \mathbf{1}[A[i] \neq B[j]] & \text{if } i, j > 0. \end{cases}$$

The edit distance between $A$ and $B$ is then captured by $D(n, n)$. When $k < n$, for all $D(i, j)$ such that $|i - j| > k$, because the length difference between $A[1 : i]$ and $B[1 : j]$ is greater than $k$, $D(i, j) > k$. Since the final answer $D(n, n) \leq k$, those positions with $|i - j| > k$ won't affect $D(n, n)$. Therefore, we only need to consider the set $\{D(i, j) : |i - j| \leq k\}$.

For $d \in [-k, k], r \in [0, k]$, we define $F(r, d) = \max_i \{i : D(i, i + d) = r\}$ and let $\text{LCP}(i, j)$ denote the length of the longest common prefix of $A[i : n]$ and $B[j : n]$. The algorithm of Landau et al. (1998) defines $\text{EXTEND}(r, d) := F(r, d) + \text{LCP}(F(r, d), F(r, d) + d)$. We have

$$F(r, d) = \max \begin{cases} \text{EXTEND}(r-1, d) + 1 & \text{if } r - 1 \geq 0; \\ \text{EXTEND}(r-1, d-1) & \text{if } d - 1 \geq -k, r - 1 \geq 0; \\ \text{EXTEND}(r-1, d+1) + 1 & \text{if } d + 1, r + 1 \leq k. \end{cases}$$

The edit distance between $A$ and $B$ equals $\min_r \{r : F(r, 0) = n\}$.

To implement LCP, Landau et al. (1998) uses a suffix tree data structure with initialization time $O(n)$ and query time $O(1)$, thus the total time complexity is $O(k^2 + n)$. In place of their suffix tree data structure, we use our DP-LCP data structure (Theorem 5.1). This leads to Algorithm 2.

**Theorem 5.3.** *Given a string $A$ of length $n$. There exists an $\epsilon$-DP data structure DPEDITDISTANCE (Algorithm 2) supporting the following operations:*

- *INIT($A \in \{0, 1\}^n$): It preprocesses an input string $A$. This procedure takes $O(n(\log k + \log \log n))$ time.*

- *QUERY($B \in \{0, 1\}^n$): For any query string $B$ with $w := D_{\text{edit}}(A, B) \leq k$, QUERY outputs a value $\widetilde{w}$ such that $|w - \widetilde{w}| \leq \widetilde{O}(k/e^{\epsilon/(\log k \log n)})$ with probability $0.99$. This procedure takes $O(n(\log k + \log \log n) + k^2 \log^2 n(\log k + \log \log n)) = \widetilde{O}(k^2 + n)$ time.*

Again, we divide the proof into runtime, privacy and utility.

## 5.1 TIME COMPLEXITY

We prove the time complexity of INIT and QUERY respectively.

**Lemma 5.4.** *The running time of INIT (Algorithm 2) is $O(n(\log k + \log \log n))$.*

*Proof.* The INIT runs DPLCP.INIT. From Theorem 5.1, the init time is $O(n(\log k + \log \log n))$. $\quad\square$

**Lemma 5.5.** QUERY *(Algorithm 2) runs in time $O((n + k^2 \log n)(\log k + \log \log n))$.*

*Proof.* The QUERY runs DPLCP.QUERYINIT once and DPLCP.QUERY $k^2$ times. From Theorem 5.1, the query time is $O(n(\log k + \log \log n) + k^2 \log^2 n(\log k + \log \log n))$. $\quad\square$

## 5.2 PRIVACY GUARANTEE

**Lemma 5.6.** *The data structure DPEDITDISTANCE (Algorithm 2) is $\epsilon$-DP.*

*Proof.* The data structure only stores a DPLCP(Algorithm 3, 4). From Theorem 5.1 and the post-processing property (Lemma 3.5), it is $\epsilon$-DP. $\quad\square$

---

**Algorithm 2** Differential Private Edit Distance

---

1: **data structure** DPEDITDISTANCE           ▷ Theorem 5.3
2: **procedure** INIT($A \in \{0,1\}^n, n \in \mathbb{N}_+, k \in \mathbb{N}_+, \epsilon \in \mathbb{R}$)     ▷ Lemma 5.4
3:    DPLCP.INIT($A, n, k, \epsilon$)           ▷ Algorithm 3
4: **end procedure**
5:
6: **procedure** EXTEND($F, i, j$)
7:    **return** $F(i,j) + \text{DPLCP.QUERY}(F(i,j), F(i,j) + j)$    ▷ Algorithm 4
8: **end procedure**
9:
10: **procedure** QUERY($B, n, k$)         ▷ Lemma 5.5 and 5.8
11:    DPLCP.QUERYINIT($B, n, k$)         ▷ Algorithm 3
12:    $F_{0,0} \leftarrow 0$
13:    **for** $i$ from 1 to $k$ **do**
14:      **for** $j \in [-k, k]$ **do**
15:        $F_{i,j} \leftarrow \max(F_{i,j}, \text{EXTEND}(i-1, j))$     ▷ Algorithm 4
16:        **if** $j - 1 \geq -k$ **then**
17:          $F_{i,j} \leftarrow \max(F_{i,j}, \text{EXTEND}(i-1, j-1))$    ▷ Algorithm 4
18:        **end if**
19:        **if** $j + 1 \leq k$ **then**
20:          $F_{i,j} \leftarrow \max(F_{i,j}, \text{EXTEND}(i-1, j+1))$    ▷ Algorithm 4
21:        **end if**
22:      **end for**
23:      **if** $F_{i,0} = n$ **then**
24:        **return** $i$
25:      **end if**
26:    **end for**
27: **end procedure**
28: **end data structure**

---

### 5.3 UTILITY GUARANTEE

Before analyzing the error of the output of QUERY (Algorithm 2), we first introduce a lemma:

**Lemma 5.7.** *Let $A, B$ be two strings. Let $\text{LCP}(i, d)$ be the length of the true longest common prefix of $A[i : n]$ and $B[i + d : n]$. For $i_1 \leq i_2, d \in [-k, k]$, we have $i_1 + \text{LCP}(i_1, d) \leq i_2 + \text{LCP}(i_2, d)$.*

*Proof.* Let $w_1 = \text{LCP}(i_1, d), w_2 = \text{LCP}(i_2, d)$. Then for $j \in [i_1, i_1 + w_1 - 1]$, $A[j] = B[j + d]$. On the other side, $w_2$ is the length of the longest common prefix for $A[i_2 : n]$ and $B[i_2 + d : n]$. So $A[i_2 + w_2] \neq B[i_2 + w_2 + d]$. Therefore, $(i_2 + w_2) \notin [i_1, i_1 + w_1 - 1]$. Since $i_2 + w_2 \geq i_2 \geq i_1$, we have $i_2 + w_2 \geq i_1 + w_1$.               $\square$

**Lemma 5.8.** *Let $\widetilde{r}$ be the output of QUERY (Algorithm 2), $r$ be the true edit distance $D_{\text{edit}}(A, B)$. With probability 0.99, we have $|r - \widetilde{r}| \leq O(k(\log k + \log \log n)/(1 + e^{\epsilon/(\log k \log n)}))$.*

*Proof.* We divide the proof into two parts. In part one, we prove that with probability 0.99, $\widetilde{r} \leq r$. In part two, we prove that with probability 0.99, $\widetilde{r} \geq r - O(k(\log k + \log \log n)/(1 + e^{\epsilon/(\log k \log n)}))$. In Theorem 5.1, with probability $1 - 1/(300k^2)$, DPLCP.QUERY satisfies two conditions. Our following discussion supposes all DPLCP.QUERY satisfies the two conditions. There are $3k^2$ LCP queries, by union bound, the probability is at least 0.99.

**Part I.** Suppose without differential privacy guarantee(using original LCP function instead of our DPLCP data structure), the dynamic programming method outputs the true edit distance. We define $F'_{i,j}$ as the dynamic programming array $F$ without privacy guarantee, EXTEND'$(i, j)$ be the result of EXTEND$(i, j)$ without privacy guarantee. Then we prove that for all $i \in [0, k], j \in [-k, k]$, $F_{i,j} \geq F'_{i,j}$ holds true.

We prove the statement above by math induction on $i$. For $i = 0$, $F(0,0) = F'(0,0) = 0$. Suppose for $i-1$, $F(i-1,j) \geq F'(i-1,j)$, then for $i$,

$$F(i,j) = \max \begin{cases} \text{EXTEND}(i-1,j)+1 & \text{if } i-1 \geq 0; \\ \text{EXTEND}(i-1,j-1) & \text{if } j-1 \geq -k, i-1 \geq 0; \\ \text{EXTEND}(i-1,j+1)+1 & \text{if } j+1, i+1 \leq k, i-1 \geq 0. \end{cases}$$

For $\text{EXTEND}(i-1,j)$, we have

$$\begin{aligned} \text{EXTEND}(i-1,j) =& F(i-1,j) + \text{DPLCP.QUERY}(F(i-1,j), F(i-1,j)+j) \\ \geq& F(i-1,j) + \text{LCP}(F(i-1,j), F(i-1,j)+j) \\ \geq& F'(i-1,j) + \text{LCP}(F'(i-1,j), F'(i-1,j)+j) \\ =& \text{EXTEND'}(i-1,j) \end{aligned}$$

The second step is because in QUERY (Theorem 5.1), $\widetilde{w} \geq w$. The third step follows from $F(i-1,j) \geq F'(i-1,j)$ and Lemma 5.7. Thus, $F(i,j) = \max_{j_2 \in [j,j-1,j+1]}\{\text{EXTEND}(i,j_2)\} \geq \max_{j_2 \in [j,j-1,j+1]}\{\text{EXTEND'}(i,j_2)\} = F'(i,j)$. Since $\widetilde{r} = \min\{\widetilde{r} : F(\widetilde{r},0) = n\}$, $r = \min\{r : F'(r,0) = n\}$, we have $F(r,0) \geq F'(r,0) = n$. Therefore $\widetilde{r} \leq r$.

**Part II.** Let $G(L,R,j) := D_{\text{edit}}(A[L:R], B[L+j, R+j])$. In this part, we prove that the edit distance $G(1,F_{i,j},j) \leq i \cdot (1 + O((\log k + \log\log n)/(1 + e^{\epsilon/(\log k \log n)})))$ by induction on $i$.

For $i = 0$, $F_{0,0} = 0$. The statement holds true. Suppose for $i-1$, $G(1,F_{i-1,j},j) \leq (i-1) \cdot (1 + O((\log k + \log\log n)/(1 + e^{\epsilon/(\log k \log n)})))$, then we prove this holds for $i$.

Because $F(i,j) = \max_{j_2 \in [j,j-1,j+1]}\{\text{EXTEND}(i,j_2)\}$, there is some $j_2 \in \{j, j-1, j+1\}$ such that $F_{i,j} = F_{i-1,j_2} + \text{DPLCP.QUERY}(F_{i-1,j_2}, F_{i-1,j_2} + j_2)$. Let $Q := \text{DPLCP.QUERY}(F_{i-1,j_2}, F_{i-1,j_2} + j_2)$. Therefore

$$\begin{aligned} G(1,F_{i,j},j) \leq& G(1, F_{i-1,j_2} + Q, j_2) + 1 \\ \leq& G(1, F_{i-1,j_2}, j_2) + G(F_{i-1,j_2}, F_{i-1,j_2} + Q, j_2) + 1 \\ \leq& G(1, F_{i-1,j_2}, j_2) + 1 + O((\log k + \log\log n)/(1 + e^{\epsilon/(\log k \log n)})) \\ \leq& i \cdot (1 + O((\log k + \log\log n)/(1 + e^{\epsilon/(\log k \log n)}))) \end{aligned}$$

The third step follows from Theorem 5.1, and the fourth step follows from the induction hypothesis. Therefore, $r = G(1, F_{\widetilde{r},0}, 0) \leq \widetilde{r} \cdot (1 + O((\log k + \log\log n)/(1 + e^{\epsilon/(\log k \log n)})))$ and the proof is completed. $\square$

**Remark 5.9.** *To the best of our knowledge, this is the first edit distance algorithm, based on* noisy *LCP implementations. In particular, we prove a structural result: if the LCP has query (additive) error $\delta$, then we could implement an edit distance data structure with (additive) error $O(k\delta)$. Compared to standard relative error approximation, additive error approximation for edit distance is relatively less explored (see e.g., Bringmann et al. (2022) for using additive approximation to solve gap edit distance problem). We hope this structural result sheds light on additive error edit distance algorithms.*

# 6 CONCLUSION

We study the differentially private Hamming distance and edit distance data structure problem in the function release communication model. This type of data structures are $\epsilon$-DP against any sequence of queries of arbitrary length. For Hamming distance, our data structure has query time $\widetilde{O}(mk + n)$ and error $\widetilde{O}(k/e^{\epsilon/\log k})$. For edit distance, our data structure has query time $\widetilde{O}(mk^2 + n)$ and error $\widetilde{O}(k/e^{\epsilon/(\log k \log n)})$. While the runtime of our data structures (especially edit distance) is nearly-optimal, it is interesting to design data structures with better utility in this model.

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

# A   PROOFS FOR HAMMING DISTANCE DATA STRUCTURE

In this section, we include all proof details in Section 4.

## A.1   PROOF OF LEMMA 4.3

*Proof of Lemma 4.3.* Let $E(A), E(A')$ be ENCODE$(A)$ and ENCODE$(A')$. Let $\#(E(A) = S)$ be the number of the same bits between $E(A)$ and $S$, $\#(E(A) \neq S)$ be the number of the different bits between $E(A)$ and $S$. Then the probability that the random flip process transforms $E(A)$ into $S$ is:

$$\Pr[\mathcal{A}(A) = S] = (\frac{1}{1 + e^{\epsilon/(2M_1)}})^{\#(E(A) \neq S)}(\frac{e^{\epsilon/(2M_1)}}{1 + e^{\epsilon/(2M_1)}})^{\#(E(A)=S)}$$
$$= \frac{(e^{\epsilon/(2M_1)})^{\#(E(A)=S)}}{(1 + e^{\epsilon/(2M_1)})^n}$$

Since for each position, ENCODE changes at most $M_1$ bits, and $A$ and $A'$ only have one different position. Therefore there are at most $2M_1$ different bits between $E(A)$ and $E(A')$. So we have

$$\frac{\Pr[\mathcal{A}(A) = S]}{\Pr[\mathcal{A}(A') = S]} \leq (e^{\epsilon/(2M_1)})^{|\#(E(A)=S)-\#(E(A')=S)|}$$
$$\leq (e^{\epsilon/(2M_1)})^{2M_1}$$
$$= e^{\epsilon}$$

Thus we complete the proof. □

## A.2   PROOF OF LEMMA 4.4

*Proof of Lemma 4.4.* $h$ is a hash function randomly drawn from all functions $[2n] \rightarrow [M_2]$. For certain $j$, $h(p) = j$ for all $p$ are independent random variables, each of them equals 1 with probability $1/M_2$, or 0 with probability $1 - 1/M_2$. So we have

$$\Pr[|T_j| \geq 10 \log k] = \Pr[\sum_{p \in T}[h(p) = j] \geq 10 \log k]$$
$$= \sum_{d=10 \log k}^{|T|} \binom{|T|}{d}(\frac{1}{M_2})^d(1 - \frac{1}{M_2})^{|T|-d}$$
$$\leq \sum_{d=10 \log k}^{|T|} \frac{|T|!}{d!(|T| - d)!}(\frac{1}{M_2})^d$$
$$\leq \sum_{d=10 \log k}^{|T|} \frac{|T|^d}{d!}(\frac{1}{M_2})^d$$
$$\leq \sum_{d=10 \log k}^{|T|} \frac{1}{d!}(\frac{1}{2})^d$$
$$\leq \frac{1}{(10 \log k)!} \sum_{d=10 \log k}^{|T|} (\frac{1}{2})^d$$
$$\leq \frac{1}{200k}$$

The fifth step follows from that fact that $|T| \leq k$, $M_2 = 2k$.

Therefore, by union bound over all $j \in [M_2]$, we can show

$$\Pr[\forall j \in [M_2], |T_j| < 10 \log k] \geq 1 - 2k \cdot (\frac{1}{200k})$$
$$= 0.99.$$

Thus, we complete the proof. □

### A.3 PROOF OF LEMMA 4.5

*Proof of Lemma 4.5.* $g$ is a hash function randomly drawn from all functions $[2n] \times [M_1] \to [M_3]$. For every single $i \in [M_1]$, define event $E_i$ as the event that the $2|T_j|$ values in $\{g(2(p-1) + A_p, i) \mid p \in T_j\} \bigcup \{g(2(p-1) + B_p, i) \mid p \in T_j\}$ are mapped into distinct positions.

$$
\begin{aligned}
\Pr[E_i] &= \prod_{c=1}^{2|T_j|} \left(1 - \frac{c}{M_3}\right) \\
&\geq 1 - \sum_{c=1}^{2|T_j|} \frac{c}{M_3} \\
&= 1 - \frac{2|T_j|(|T_j + 1|)}{M_3} \\
&> 1 - \frac{2(10 \log^2 k)}{400 \log^k} \\
&= 0.5
\end{aligned}
$$

The fourth step follows from Lemma 4.4. It holds true with probability 0.99.

For different $i \in [M_1]$, $E_i$ are independent. Therefore, the probability that all $E_i$ are false is $(0.5)^{M_1} < 1/(1000k)$. By union bound, the probability that for every $j \in [M_2]$ there exists at least one $i$ such that $E_i$ is true is at least

$$
1 - M_2 \cdot 0.5^{M_1} \geq 1 - M_2/(1000k) \geq 0.98. \qquad \square
$$

### A.4 PROOF OF LEMMA 4.6

*Proof of Lemma 4.6.* From Lemma 4.5, for all $j$, there is at least one $i$, such that the set $\{g(2(p-1) + A_p, i) \mid p \in T_j\} \bigcup \{g(2(p-1) + B_p, i) \mid p \in T_j\}$ contains $2|T_j|$ distinct values. Therefore, for that $i$, $E(A)_{i,j,1 \sim M_3}$ and $E(B)_{i,j,1 \sim M_3}$ have exactly $2|T_j|$ different bits. For the rest of $i$, the different bits of $E(A)_{i,j,1 \sim M_3}$ and $E(B)_{i,j,1 \sim M_3}$ is no more than $2|T_j|$. So we have $0.5 \cdot \sum_{j=1}^{M_2} \max_{i \in [M_1]} (\sum_{c=1}^{M_3} |E(A)_{i,j,c} - E(B)_{i,j,c}|) = 0.5 \cdot \sum_{j=1}^{M_2} 2|T_j| = |T| = D_{\text{ham}}(A, B).$ $\qquad \square$

## B DIFFERENTIALL PRIVATE LONGEST COMMON PREFIX

We design an efficient, $\epsilon$-DP longest common prefix (LCP) data structure in this section. Specifically, for two positions $i$ and $j$ in $A$ and $B$ respectively, we need to calculate the maximum $l$, so that $A[i : i + l] = B[j : j + l]$. For this problem, we build a differentially private data structure (Algorithm 3 and Algorithm 4). The main contribution is a novel utility analysis that accounts for the error incurred by differentially private bit flipping.

### B.1 TIME COMPLEXITY

We prove the running time of the three operations above.

**Lemma B.1.** *The running time of* INIT *and* INITQUERY *(Algorithm 3) are* $O(n \log n(\log k + \log \log n))$

*Proof.* From Lemma 4.2, the running time of building node $T_{i,j}$ is $O((n/2^i)M_1)$. Therefore the total building time of all nodes is

$$
\sum_{i=0}^{\log n} \sum_{j=0}^{2^i - 1} (n/2^i)M_1 = \sum_{i=0}^{\log n} 2^i \cdot (n/2^i)M_1 = O(n \log n(\log k + \log \log n)).
$$

Thus, we complete the proof. $\qquad \square$

**Lemma B.2.** *The running time of* QUERY *(Algorithm 4) is* $O(\log^2 n(\log k + \log \log n))$.

---

**Algorithm 3** Differential Private Longest Common Prefix, Part 1

---

1: **data structure** DPLCP                                                                    ▷ Theorem 5.1
2: **members**
3:    $T_{i,j}^A, T_{i,j}^B$ for all $i \in [\log n], j \in [2^i]$
4:       ▷ $T_{i,j}$ represents the hamming sketch (Algorithm 1) of the interval $[i \cdot n/2^j, (i+1) \cdot n/2^j]$
5: **end members**
6:
7: **procedure** BUILDTREE($A \in \{0,1\}^n, n \in \mathbb{N}_+, k \in \mathbb{N}_+, \epsilon \in \mathbb{R}$)                    ▷ Lemma B.3
8:    $M_1 \leftarrow \log k + \log \log n + 10, M_2 \leftarrow 1, M_3 \leftarrow 10, \epsilon' \leftarrow \epsilon/\log n$
9:    **for** $i$ from 0 to $\log n$ **do**
10:       **for** $j$ from 0 to $2^i - 1$ **do**
11:          $T_{i,j}^* \leftarrow$ DPHAMMINGDISTANCE.INIT($A[j \cdot n/2^i : (j+1) \cdot n/2^i], M_1, M_2, M_3, \epsilon'$)
12:                                                                                                ▷ Algorithm 1
13:       **end for**
14:    **end for**
15:    **return** $T^*$
16: **end procedure**
17:
18: **procedure** INIT($A \in \{0,1\}^n, n \in \mathbb{N}_+, k \in \mathbb{N}_+, \epsilon \in \mathbb{R}$)                    ▷ Lemma B.1
19:    $T^A \leftarrow$ BUILDTREE($A, n, k, \epsilon$)
20: **end procedure**
21:
22: **procedure** QUERYINIT($B \in \{0,1\}^n, n \in \mathbb{N}_+, k \in \mathbb{N}_+$)                    ▷ Lemma B.1
23:    $T^B \leftarrow$ BUILDTREE($B, n, k, 0$)
24: **end procedure**
25:
26: **procedure** INTERVALSKETCH($T, p_l \in [n], p_r \in [n]$)
27:    Divide the interval $[p_l, p_r]$ into $O(\log n)$ intervals. Each of them is stored on a node of the tree $T$.
28:    Retrieve the Hamming distance sketches of these nodes as $S_1, S_2, \ldots, S_t$.
29:    Initialize a new sketch $S \leftarrow 0$ with the same size of the sketches above.
30:    **for** every position $w$ in the sketch $S$ **do**
31:       $S[w] \leftarrow S_1[w] \oplus S_2[w] \oplus S_3[w] \oplus \ldots \oplus S_t[w]$
32:    **end for**
33:    **return** $S$
34: **end procedure**
35:
36: **procedure** SKETCHHAMMINGDISTANCE($S^A, S^B \in \mathbb{R}^{M_1 \times M_2 \times M_3}$)       ▷ Lemma B.4 and B.5
37:    Let $M_1, M_2, M_3$ be the size of dimensions of the sketches $S^A$ and $S^B$.
38:    **return** $0.5 \cdot \sum_{j=1}^{M_2} \max_{i \in [M_1]} (\sum_{c=1}^{M_3} |S_{i,j,c}^A - S_{i,j,c}^B|)$
39: **end procedure**
40: **end data structure**

---

*Proof.* In QUERY (Algorithm 4), we use binary search. There are totally $\log n$ checks. In each check, we need to divide the interval into $\log n$ intervals and merge their sketches of size $M_1 M_2 M_3$. So the time complexity is $O(\log^2 n(\log k + \log \log n))$.                                                     □

### B.2    PRIVACY GUARANTEE

**Lemma B.3.** *The data structure* DPLCP *(Algorithm 3 and Algorithm 4) is $\epsilon$-DP.*

*Proof.* On each node, we build a hamming distance data structure DPHAMMINGDISTANCE that is $(\epsilon/\log n)$-DP. For two strings $A$ and $A'$ that differ on only one bit, since every position is in at most $\log n$ nodes on the tree, for any output $S$, the probability

$$\frac{\Pr[\text{BUILDTREE}(A) = S]}{\Pr[\text{BUILDTREE}(A') = S]} \leq (e^{\epsilon/\log n})^{\log n} = e^{\epsilon}$$

---

**Algorithm 4** Differential Private Longest Common Prefix, Part 2

---

1: **data structure** DPCLP                                              ▷ Theorem 5.1
2: **procedure** QUERY($i \in [n], j \in [n]$)                           ▷ Lemma B.2 and B.6
3:    $L \leftarrow 0, R \leftarrow n$
4:    **while** $L \neq R$ **do**
5:        $\text{mid} \leftarrow \lceil \frac{L+R}{2} \rceil$
6:        $S^A \leftarrow \text{INTERVALSKETCH}(T^A, i, i + \text{mid})$     ▷ Algorithm 3
7:        $S^B \leftarrow \text{INTERVALSKETCH}(T^B, j, j + \text{mid})$     ▷ Algorithm 3
8:        $\text{threshold} \leftarrow 1.5 M_1 M_3 / (1 + e^{\epsilon/(\log k \log n)})$
9:        **if** $\text{SKETCHHAMMINGDISTANCE}(S^A, S^B) \leq \text{threshold}$ **then**     ▷ Algorithm 3
10:          $L \leftarrow \text{mid}$
11:        **else**
12:          $R \leftarrow \text{mid} - 1$
13:        **end if**
14:    **end while**
15:    **return** $L$
16: **end procedure**
17: **end data structure**

---

Thus we complete the proof.        $\square$

### B.3   UTILITY GUARANTEE

Before analyzing the error of the query, we first bound the error of SKETCHHAMMINGDISTANCE (Algorithm 3).

**Lemma B.4.** *We select $M_1 = \log k + \log \log n + 10, M_2 = 1, M_3 = 10$ for* DPHAMMINGDISTANCE *data structure in* BUILDTREE*(Algorithm 3). Let $z$ be the true hamming distance of the two strings $A[i : i + \text{mid}]$ and $B[i : i + \text{mid}]$. Let $\widetilde{z}$ be the output of* SKETCHHAMMINGDISTANCE*(Algorithm 3). When $\epsilon = +\infty$(without the random flip process), then we have*

- *if $z = 0$, then with probability 1, $\widetilde{z} = 0$.*

- *if $z \neq 0$, then with probability $1 - 1/(300k^2 \log n)$, $\widetilde{z} \neq 0$.*

*Proof.* Our proof follows from the proof of Lemma 4.6. We prove the case of $z = 0$ and $z \neq 0$ respectively.

When $z = 0$, it means the string $A[i : i + \text{mid}]$ and $B[i : i + \text{mid}]$ are identical. Therefore, the output of the hash function is also the same. Therefore, the output $S^A$ and $S^B$ from INTERVALSKETCH(Algorithm 3) are identical. Then $\widetilde{z} = 0$.

When $z \neq 0$, define set $Q := \{p \in [\text{mid}] \mid A[i + p - 1] \neq B[j + p - 1]\}$ as the positions where string $A$ and $B$ are different. $|Q| = z$. Note that $M_1 = \log k + \log \log n + 10, M_2 = 1, M_3 = 10$, $S^A, S^B \in \{0, 1\}^{M_1 \times M_2 \times M_3}$. For every $i' \in [M_1]$, the probability that $S_{i'}^A$ and $S_{i'}^B$ are identical is the probability that all $c \in [M_3]$ is mapped exactly even times from the position set $Q$. Formally, define event $E$ as $[\forall j', |\{p \in Q \mid g(p) = j'\}| \bmod 2 = 0]$. Define another event $E'$ as there is only one position mapped odd times from set $Q_{1 \sim z-1}$. Then the probability equals

$$\Pr[E] = \Pr[E'] \cdot \Pr[E|E']$$
$$\leq \Pr[E|E']$$
$$= 1/M_3$$

The last step is because $\Pr[E|E']$ is the probability that $g(Q_z)$ equals the only position that mapped odd times. There are totally $M_3$ positions and the hash function $g$ is uniformly distributed, so the probability is $1/M_3$.

For different $i' \in [M_1]$, the event $E$ are independent. So the total probability that $z' \neq 0$ is the probability that for at least one $i'$, event $E$ holds true. So the probability is $1 - (1/M_3)^{M_1} = 1 - (1/10)^{\log k + \log \log n + 10} > 1 - 1/(300k^2 \log n)$.    $\square$

**Lemma B.5.** *Let $M_1 = \log k + \log \log n + 10, M_2 = 1, M_3 = 10$. Let $z$ be the true hamming distance of the two strings $A[i : i + \text{mid}]$ and $B[i : i + \text{mid}]$. Let $\widetilde{z}$ be the output of* SKETCHHAM-MINGDISTANCE*(Algorithm 3). With the random flip process with DP parameter $\epsilon$, we have:*

- *When $z = 0$, with probability $1 - 1/(300k^2 \log n)$, $\widetilde{z} < (1 + o(1))M_1 M_3/(1 + e^{\epsilon/(\log k \log n)})$.*

- *When $z > 3M_1 M_3/(1 + e^{\epsilon/(\log k \log n)})$, with probability $1 - 1/(300k^2 \log n)$, $\widetilde{z} > (2 - o(1))M_1 M_3/(1 + e^{\epsilon/(\log k \log n)})$.*

*Proof.* In the random flip process in DPHAMMINGDISTANCE (Algorithm 1,3), the privacy parameter $\epsilon' = \epsilon/\log n$. We flip each bit of the sketch with independent probability $1/(1 + e^{\epsilon/\log n \log k})$. Then we prove the case of $z = 0$ and $z > 3M_1 M_3/(1 + e^{\epsilon/(\log k \log n)})$ respectively.

When $z = 0$, similar to the proof of Lemma 4.7, we view the flipping operation as random variables. Let random variable $R_{i,j,c}$ be 1 if the sketch $S_{i,j,c}^A$ is flipped, otherwise 0. From Lemma B.4, $S^A$ and $S^B$ are identical. Then we have

$$|z - \widetilde{z}| = \max_{i \in [M_1]} \sum_{c=1}^{M_3} R_{i,j,c}$$

$$\leq \sum_{i=1}^{M_1} \sum_{c=1}^{M_3} R_{i,j,c}$$

Since $R_{i,j,c}$ are independent Bernoulli random variables, using Hoeffding's inequality (Lemma 3.2), we have

$$\Pr[|\sum_{i=1}^{M_1 \times M_3} R_{i,j,c} - M_1 M_3 \, \mathbb{E}[R_{i,j,c}]| > L] \leq e^{-2L^2/(M_1 \times M_3)}$$

When $L = M_1 \sqrt{M_3}$,

$$\Pr[|\sum_{i=1}^{M_1 \times M_3} R_{i,j,c} - M_1 M_3 \, \mathbb{E}[R_{i,j,c}]| > L] \leq e^{-2M_1^2 M_3/(M_1 M_3)}$$

$$\leq e^{-2(\log k + \log \log n)}$$

$$\leq 1/(300k^2 \log n)$$

Thus we complete the $z = 0$ case.

When $z > 3M_1 M_3/(1 + e^{\epsilon/(\log k \log n)})$, the proof is similar to $z = 0$. With probability $1 - 1/(300k^2 \log n)$, we have $|z - \widetilde{z}| < (1 + o(1))M_1 M_3/(1 + e^{\epsilon/(\log k \log n)})$. Thus $\widetilde{z} > (2 - o(1))M_1 M_3/(1 + e^{\epsilon/(\log k \log n)})$. $\qquad\square$

**Lemma B.6.** *Let $\widetilde{w}$ be the output of* QUERY$(i, j)$ *(Algorithm 4), $w$ be the longest common prefix of $A[i : n]$ and $B[j : n]$. With probability $1 - 1/(300k^2)$, we have: 1.$\widetilde{w} \geq w$. 2. $D_{\text{ham}}(A[i : i + \widetilde{w}], B[j : j + \widetilde{w}]) \leq 3M_1 M_3/(1 + e^{\epsilon/(\log k \log n)})$.*

*Proof.* In QUERY$(i, j)$ (Algorithm 4), we use a binary search to find the optimal $w$. In binary search, there are totally $\log n$ calculations of SKETCHHAMMINGDISTANCE. Define $\text{threshold} := 1.5M_1 M_3/(1 + e^{\epsilon/(\log k \log n)})$. Define a return value of SKETCHHAMMINGDISTANCE is good if: 1). when $z = 0$, $\widetilde{z} < \text{threshold}$. 2). when $z > 2 \cdot \text{threshold}$, $\widetilde{z} < \text{threshold}$. $z$ and $\widetilde{z}$ are defined in Lemma B.5.

Therefore, by Lemma B.5, each SKETCHHAMMINGDISTANCE is good with probability at least $1 - 1/(k^2 \log n)$. There are $\log n$ SKETCHHAMMINGDISTANCE in the binary search, by union bound, the probability that all of them are good is at least $1 - 1/(300k^2)$.

When all answers for SKETCHHAMMINGDISTANCE are good, from the definition of binary search, for any two positions $L, R$ such that $D_{\text{ham}}(A[i : i + L], B[j, j + L]) = 0$, $D_{\text{ham}}(A[i : i + R], B[j, j +$

$R]) \geq 2 \cdot \text{threshold}$, we have $L \leq \widetilde{w} \leq R$. Next, we prove $w \leq \widetilde{w}$ and $D_{\text{ham}}(A[i : i + \widetilde{w}], B[j : j + \widetilde{w}]) \leq 3M_1 M_3 / (1 + e^{\epsilon / (\log k \log n)})$ respectively.

$w$ is the true longest common prefix of $A[i : n]$ and $B[j : n]$, so we have $D_{\text{ham}}(A[i : i + L], B[j, j + L]) = 0$. Let $L = w$, we have $w = L \leq \widetilde{w}$.

Let $R$ be the minimum value that $D_{\text{ham}}(A[i : i + R], B[j : j + R]) \geq 2 \cdot \text{threshold}$. Because $D_{\text{ham}}(A[i : i + R], B[j, j + R])$ is monotone for $R$, and $\widetilde{w} \leq R$, we have $D_{\text{ham}}(A[i : i + \widetilde{w}], B[j : j + \widetilde{w}]) \leq D_{\text{ham}}(A[i : i + R], B[j : j + R]) = 2 \cdot \text{threshold} = 3M_1 M_3 / (1 + e^{\epsilon / (\log k \log n)})$.

Thus we complete the proof. $\qquad\qquad\qquad\qquad\qquad\qquad\qquad\qquad\qquad\qquad\qquad\qquad\quad\square$

