# OpenReview forum: "On Differentially Private String Distances"
_ICLR.cc/2025/Conference — ICLR 2025 Conference Withdrawn Submission_

### Official Review · Reviewer_gzSh · 2024-11-01

**Soundness:** 2
**Presentation:** 2
**Contribution:** 2
**Rating:** 3
**Confidence:** 3

**Summary:**

This paper studies the problem of privately computing data structures for answering binary string distance queries: given a collection of $m$ length-$n$ binary vectors, compute a data structure that is $\varepsilon$-DP with respect to the vectors and also approximates string distance between each vector and any arbitrary query vector. The paper provides solutions for both Hamming and edit distance that use time and space (roughly) linear in $m$ and $n$, with accompanying utility guarantees.

**Strengths:**

Private string edit distance is a natural problem, and the paper provides a thorough treatment of its algorithms, in the sense of analyzing privacy, time, space, and utility. I'm not aware of relevant work in differential privacy (https://arxiv.org/abs/2311.07415 might be closest, but I don't think it directly applies here).

**Weaknesses:**

1. I'm confused by the utility guarantees for these algorithms. As stated in Theorem 1.1, for any query string within Hamming distance k of the database strings, it estimates these distances to additive error $\tilde O(k / e^{\varepsilon / \log(k)})$. For the typical setting of small $\varepsilon$, isn't this essentially $\tilde O(k)$ error, which would also be achieved by the trivial algorithm that estimates each distance as 0? How large an $\varepsilon$ is necessary to beat the trivial algorithm? The same question applies for edit distance. I think I must be misunderstanding something here, because it makes the utility guarantees for these algorithms look trivial.

2. The algorithmic novelty for the Hamming distance data structure is unclear. It is described as an adaptation of a non-private approach from the Porat and Lipsky paper -- is it just that algorithm + randomized response, or are other modifications necessary? I understand that the paper positions the main "work" here as the utility analysis, but if there's more algorithmic novelty, it would be good to highlight it.

3. For boosting, we'd need to pay a privacy cost for each data structure copy. Is there any reason to believe this would actually improve accuracy (for a fixed privacy budget)? That is, if $\varepsilon$ is fixed, is there any reason to use boosting (and divide $\varepsilon$ among the copies)?

4. The paper doesn't attempt to explain the core privatization algorithm for the edit distance problem in the main body, instead spending several pages analyzing algorithms that rely on it. Since the overall privacy guarantee relies on this algorithm, omitting it from the main paper seems odd. More generally, I think the paper is missing sketches of how the algorithms work (maybe because the ideas are simple/familiar to people who work on string distances -- but not to me). Dumping blocks of pseudocode and moving on technically compiles, but it doesn't make for a readable paper.

5. The paper should be clarify its neighboring relation in something more prominent than a footnote, especially since the neighboring relation is more restrictive (change one position in one string) than I expected from the introduction's example (change one string).

Overall, the problems are natural, but I'm not sure what the primary contribution here is supposed to be.

**Questions:**

(See Weaknesses section.)

---

### Official Review · Reviewer_UBbX · 2024-11-01

**Soundness:** 3
**Presentation:** 2
**Contribution:** 1
**Rating:** 3
**Confidence:** 3

**Summary:**

This paper studies differentially privately approximating the bit strings in a database, which is a fundamental problem in data structure. In particular, given datasets $A_1, A_2, \cdots, A_m\in \lbrace 0,1\rbrace^n$, the task is to output a sketch $\hat{e}(\cdot)$ with privacy, which could be used to answer any string query $B\in  \lbrace 0,1\rbrace^n$. Answering each query is private due to the robustness of differential privacy for post-processing.

The authors consider two different natural error metric: Hamming and edit distance. Under the assumption that $||A_i - B||_1 \leq k$ for any $i\in [m]$, for Hamming distance, they provide an algorithm that outputs a private synthetic sketch in $O(mn)$ time, and processes each query in time only $O(mk+n)$. The error inccurs for each query is $O(k\log k /e^{\epsilon/\log k})$. To achieve this, they privatize the algorithm in Porat & Lipsky (2007) by filpping each coordinate according to a carefully chosen probability in the sketch. The results and techniques for edit distance metric is similar.

**Strengths:**

1. The non-private version of this problem is one of the fundamental tasks in data structure.
2. Most theorems and lemmas are clearly stated.

**Weaknesses:**

I would suggest rejecting this paper due to its weakness in both results and techniques.

- For the results, this paper show that when $||A_i - B||_1 \leq k$, the error on approximating Hanmming distance between each $A_i$ and $B$ is $O(k\log k)$ for any constant $\epsilon$, if I understand it correctly. This looks trivial to me. I understand that the error can be reduced by amplifying $\epsilon$, but it is still pretty weak, as it offers little advantage within the most common range of $\epsilon$ over applying the Laplace mechanism to add noise to each coordinate.
- The authors claim that they present a "novel scheme" that flip each bit with certain probability. But to me it is just randomized response, so I am not sure if I agree this is "novel". Besides, it appears to me that Algorithm 1 closely follows the sketching algorithm in Porat & Lipsky (2007).

However, it is possible that I misunderstand the importance of the results or techniques, so I am open to discuss.

**Questions:**

I have no questions.

---

### Official Review · Reviewer_pMzp · 2024-11-04

**Soundness:** 4
**Presentation:** 3
**Contribution:** 3
**Rating:** 6
**Confidence:** 4

**Summary:**

The paper proposes data structures for computing the Hamming or edit distance of a query string to a given string, assuming the distances are promised to be at most k. The data structures satisfy differential privacy when a single character of the given string is changed. The Hamming distance algorithm adds a random flip step once a standard sketching technique is used. The more involved edit distance algorithm uses the fact that edit distance may be computed using a least common prefix data structure, which may be approximated with a set of private Hamming distance data structures for range queries.

**Strengths:**

The algorithms are a novel improvement to string algorithms under differential privacy, for which little was known before. The algorithms are rather simple, have efficient running times, and are analyzed elegantly and tightly.

**Weaknesses:**

The notion of adjacency used in the privacy guarantee is a bit weak; it applies when a single character of the input set of strings is changed. If each string corresponds to a user (e.g. DNA or utterances), this seems like an inadequate level of privacy for each user.

A second weakness is that the paper requires epsilon to be at least log k (resp. log k log n) in order to obtain non trivial utility guarantees for the Hamming distance (resp. edit distance). These values of epsilon are much better than naive approaches, but are still super-constant; and log n in particular may be too large.

**Questions:**

What are some real-world examples where the adjacency notion makes sense?

The max distance of k assumption seems more motivated for edit distance, in applications like DNA. Can you provide more motivation for this assumption on the Hamming distance?

In my opinion, the main contribution of the paper is the edit distance algorithm, of which the LCP is a major subroutine (yet is put in the appendix). Consider shortening sections 3 and 4, and presenting the Hamming distance and LCP algorithms in their own section as “building blocks” towards the main section describing the edit distance algorithm.

The relaxation from a max to a sum in line 323 seems wasteful. Is it possible to do a tighter analysis and shave off a log factor?

---

### Official Review · Reviewer_hRHc · 2024-11-04

**Soundness:** 2
**Presentation:** 3
**Contribution:** 2
**Rating:** 5
**Confidence:** 3

**Summary:**

The paper studies the problem of estimating distances between a given set of strings and the query string. The distance metric used in this paper is hamming distance and edit distance. For both of these metrics, they give a low-space algorithm that has an error that scales with $O(k/e^{\epsilon/\log(k)})$ and $O(k/e^{\epsilon/\log(k)\log(n)})$, respectively.

**Strengths:**

The bound in the error scales exponentially with $\epsilon$. It is the first paper that studies data structure which provides non-trivial accuracy for measuring string distances, both with respect to the edit distance as well as the hamming distance.

The query time of the data structure is also sublinear.

**Weaknesses:**

To me, it felt like the paper is computing the sketch using known techniques and using randomized response. The rest of the analysis seems more like an analysis of the error due to the randomized response. So, I do not see much of novelty in the paper. I do think the question studied is somewhat interesting, but again I felt that the motivation to study the question and the privacy definition needs a bit more clarification. The motivation for the neighboring relation studied in the paper is a little weak.

**Questions:**

1. What real world motivation for the privacy definition considered in this paper?

2. What are the novel part of the paper? Is it mainly randomized response which we know is optimal for pure-DP.

---

### Author Response · Authors · 2024-11-26
**Withdrawal of the paper**

Dear Reviewers,
Thank you for your thoughtful feedback on our submission. Upon reflecting on the reviews, we have decided to withdraw our manuscript and make major revisions to improve it based on feedback. Your feedback will be invaluable as we improve the manuscript. We appreciate your time and efforts for providing insightful feedback for this work.

Best,
Authors

---

### Note · Authors · 2024-11-26

I have read and agree with the venue's withdrawal policy on behalf of myself and my co-authors.